# Immunolocalization of Two Neurotrophins, NGF and BDNF, in the Pancreas of the South American Sea Lion *Otaria flavescens* and Bottlenose Dolphin *Tursiops truncatus*

**DOI:** 10.3390/ani14162336

**Published:** 2024-08-13

**Authors:** Claudia Gatta, Luigi Avallone, Anna Costagliola, Paola Scocco, Livia D’Angelo, Paolo de Girolamo, Elena De Felice

**Affiliations:** 1Department of Veterinary Medicine and Animal Production, University of Naples Federico II, Via F. Delpino 1, 80137 Naples, Italy; claudia.gatta@unina.it (C.G.); luigi.avallone@unina.it (L.A.); anna.costagliola@unina.it (A.C.); livia.dangelo@unina.it (L.D.); degirola@unina.it (P.d.G.); 2School of Biosciences and Veterinary Medicine, University of Camerino, Via Gentile III da Varano, 62032 Camerino, Italy; paola.scocco@unicam.it

**Keywords:** Nerve Growth Factor, Brain-Derived Neurotrophic Factor, pancreas, common bottlenose dolphin, South American sea lion, marine mammals

## Abstract

**Simple Summary:**

In this work, we documented the presence of two neurotrophins NGF (Nerve Growth Factor) and BDNF (Brain-Derived Neurotrophic Factor) in the pancreas of two species of marine mammals, *Tursiops truncatus* (common bottlenose dolphin) and *Otaria flavescens* (South American sea lion). Pancreas samples were provided by the Mediterranean Marine Mammal Tissue Bank of the Department of Comparative Biomedicine and Food Science of the University of Padua (Italy). These two growth factors are well-known mediators for maintaining the survival of neurons but also exert a metabotrophic effect. The localization in the pancreas of bottlenose dolphins and sea lions of these neurotrophins could help to better understand how these marine animals handle metabolic challenges in their adaptation to ocean life.

**Abstract:**

In this study, we have investigated the immunolocalization of NGF (Nerve Growth Factor) and BDNF (Brain-Derived Neurotrophic Factor) in the pancreas of two species of marine mammals: *Tursiops truncatus* (common bottlenose dolphin), belonging to the order of the Artiodactyla, and *Otaria flavescens* (South American sea lion), belonging to the order of the Carnivora. Our results demonstrated a significant presence of NGF and BDNF in the pancreas of both species with a wide distribution pattern observed in the exocrine and endocrine components. We identified some differences that can be attributed to the different feeding habits of the two species, which possess a different morphological organization of the digestive system. Altogether, these preliminary observations open new perspectives on the function of neurotrophins and the adaptive mechanisms of marine mammals in the aquatic environment, suggesting potential parallels between the physiology of marine and terrestrial mammals.

## 1. Introduction

Nerve Growth Factor (NGF) and Brain-Derived Neurotrophic Factor (BDNF) along with neurotrophin (NT) 3 and NT4/5 belong to the family of neurotrophins, a family of structurally related growth factors [1,2,3,4]. Neurotrophins are secreted proteins that promote the development, survival, and function of the central and peripheral nervous system [1]. The signaling of each neurotrophin is mediated by a specific tyrosine kinase receptor of the Trk family (Trk A, Trk B, and Trk C). In addition, all neurotrophins activate the p75 neurotrophin receptor, a member of the tumor necrosis factor receptor superfamily [5,6]. Neurotrophins and their Trk receptors are phylogenetically preserved from invertebrates to mammals playing an essential role in the nervous system development and serving important functions in certain non-nervous tissues [7]. NGF is a neurotrophic factor that promotes the growth of neuritis during development and provides trophic support to sensory neurons, sympathetic neurons, and some cholinergic neurons [8,9], presents trophic actions in the cells of the endocrine and immune system, participates in acute inflammatory responses [10], and responds to stress to regulate the functions of the anterior pituitary cells [11]. Different studies have shown that NGF was implicated in the morphogenesis and ontogeny of pancreatic islets [12,13] while the integrity of the NGF/NGF receptor and NGF bioavailability participate in controlling β-cell survival in culture [14]. In adults, NGF increases glucose-stimulated insulin secretion [15]. In fact, it modifies the physiology of β-cells that synthesize and secrete NGF in response to increased extracellular glucose concentrations and this implies its autocrine/paracrine role in the pancreas [16,17]. BDNF not only serves as a powerful pro-survival factor for neurons in both the central and peripheral nervous system, as well as a strong modulator of synaptic plasticity [18], but it also plays a crucial role in the regulation of body weight, energy expenditure, and glucose metabolism [19]. When brain BDNF levels are depleted, it can lead to excessive nutrition and increased body weight, which are often accompanied by metabolic syndrome characterized by hyperleptinemia, hyperglycemia, and hyperinsulinemia [20]. BDNF is involved in β-cell survival by binding to TrkB [21]. With regard to glucose metabolism, it has been shown that plasma BDNF is inversely related to plasma glucose, which raised the possibility that elevated plasma glucose levels adversely affected BDNF production [22]. Hanyu and coworkers demonstrated that BDNF is also involved in the secretion of glucagon by the alpha cells of the pancreas [19]. Moreover, BDNF enhances insulin sensitivity in peripheral tissues, boosts energy expenditure in db/db mice, and reduces their food intake [23,24].

These two neurotrophins have been extensively studied in the pancreas of terrestrial mammals [25,26,27,28,29] as well as in birds and reptiles [30] but they have never been investigated in marine mammals.

Marine mammals, with their anatomy and physiology adapted to aquatic life [31,32], offer an interesting context to explore the distribution and role of neurotrophins in organs critical to metabolic regulation, such as the pancreas. The pancreas is a key organ in the regulation of glucose metabolism and energy homeostasis [33,34] and may provide valuable insights into the evolutionary adaptive mechanisms of marine mammals in aquatic environments.

In this study, we focused on *Tursiops truncatus* (common bottlenose dolphin), belonging to the order of Artiodactyla, and *Otaria flavescens* (South American sea lion), belonging to the order ofCarnivora. These two species have evolved different feeding habits and related morphological gastro-enteric apparatus, with peculiar metabolic adaptations [31,32]. For instance, the bottlenose dolphin has metabolic conditions that can be compared to human pre-diabetes, displaying hyperinsulinemia, hyperlipidemia, and prolonged postprandial hyperglycemia [35,36]. Despite these metabolic conditions, bottlenose dolphins do not develop ketosis during prolonged fasting periods, suggesting the existence of unique physiological adaptation mechanisms [37] and making them natural models in metabolic adaptation mechanisms in response to energy challenges. On the other hand, sea lions have the ability to maintain metabolic homeostasis and normal neurological function under extreme conditions: they face extended periods of prolonged fasting linked to their reproductive season, during which they must manage energy reserves and maintain neurological function despite the lack of food availability [38].

Over the last years, our research group has turned research interest toward the gastro-entero-pancreatic system of marine mammals by identifying key morphological traits, also in terms of immunolocalization of neuropeptides [39,40,41,42]. This short communication aims to add further comparative morphological aspects of two different marine mammal species.

## 2. Materials and Methods

Mediterranean Marine Mammal Tissue Bank (MMMTB) of the Department of Comparative Biomedicine and Food Science of the University of Padua (Italy) kindly supplied pancreas samples of common bottlenose dolphins *Tursiops truncatus* and South America sea lion *Otaria flavescens.* Ethical approval was not required for this work because tissue stored at MMMTB (CITES institution IT020) was derived from stranded animals or from marine mammals who died in captivity and were referred for postmortem.

Samples were first fixed in 10% buffered formalin, then paraffin included and finally cut into 8 µm-thick serial sections.

The sections were stained with Harris’s hematoxylin and eosin (HE) for the histological evaluation at the optic level.

The distributions of NGF and BDNF were studied by single immunohistochemistry. Briefly, the sections were dewaxed and incubated with 0.3% hydrogen peroxide for 30 min at room temperature (RT) to block endogenous peroxidase activity. Then, the sections were rinsed in 0.01 M phosphate-buffered saline (PBS), pH 7.4, for 15 min and subsequently incubated for 20 min at RT with normal goat serum (NGS, 1:5 in 0.01 M PBS) (MP biomedicals LLC, Irvine, CA, USA cat# 191356). Then, sections were incubated overnight at 4 °C with primary antisera, respectively, rabbit polyclonal antibody against NGF diluted at 1:200 (H-20, sc-548 Santa Cruz Biotechnology, Inc., Santa Cruz, CA, USA) and rabbit polyclonal antibody against BDNF diluted at 1:200 (N-20, sc-546 Santa Cruz Biotechnology, Inc., Santa Cruz, CA, USA) [43]. The next day, the sections were rinsed in PBS for 15 min and incubated for 30 min at RT with EnVision + System-HRP, Labelled Polymer anti-rabbit (Dako, Santa Cruz, CA, USA). Subsequently, the sections were rinsed in PBS for 15 min and then incubated for 30 min at RT with avidin-peroxidase complex. Peroxidase activity was detected using a solution of 3-3′ diaminobenzidine tetrahydrocloride (Sigma-Aldrich, cat# D5905, St. Louis, MO, USA) of 10 mg in 15 mL 0.5 M Tris buffer, pH 7.6, containing 0.03% hydrogen peroxide.

Positive controls were performed by sections of zebrafish taste buds [44]. Internal reaction controls were performed by replacing primary or secondary antisera with phosphate-buffered saline or normal serum in the specific phase [45]. Control images are reported in Appendix A.

All the stained sections were photographed using a Leica DMRA2 microscope. The digital raw images were optimized for contrast and illumination by using Adobe Photoshop CS5 (Adobe Systems, San Jose, CA, USA).

## 3. Results

### 3.1. Morphological Remarks

The pancreas of *Otaria flavescens* is divided into lobules separated by bands of loose fibrous connective tissue of varying thickness (connective septa), in which abundant presence of scattered fat lobules is observed. Many exocrine acini (Figure 1A) and a variable number of randomly distributed Langerhans islets are observed in the pancreatic lobules. The islets of Langerhans exhibit different morphology; they can be very large (Figure 1A) or showing medium oval-shaped (Figure 1B) or elongate shape. Within the lobules and in the connective septa, the presence of blood vessels (Figure 1C) and ganglion with neurons (Figure 1D) is observed.

The pancreas of *Tursiops truncatus* is divided into lobules separated by loose connective septa. In the pancreatic lobules, there are numerous exocrine acini and Langerhans islets of different sizes, mainly small (Figure 1E–G) and medium (Figure 1F) delimited by a thin septum of loose fibrous tissue while rarely very large islets have been found. Most of the islets exhibit an irregular contour and shape; however, a small number of islets are oval-shaped. Within the lobules and in the connective septa the presence of blood vessels is observed (Figure 1G–H).

### 3.2. Immunohistochemical Observations

NGF was distributed in the pancreas of both species. The labeling was always observed in the cytoplasmic compartment of the positive cells. In the exocrine component of *Otaria flavescens,* a strong immunoreactivity was observed in a limited number of disseminated cells in the parenchyma (Figure 2A). In the endocrine component, a strong immunoreactivity was detected in almost all the cells of the islet of Langerhans (Figure 2B). In the connective septa, the presence of positive fibers along the wall of blood vessels (Figure 2C) and positive neurons in a ganglion (Figure 2D) was observed.

In *Tursiops truncatus*, a moderate NGF signal was seen in disseminated cells in the exocrine component (Figure 2E). In the endocrine component, NGF immunoreactivity was observed in a few islets of Langerhans (Figure 2F). Immunoreactivity to NGF was also found in fibers along the walls of the blood vessels (Figure 2G) and fibers along the connective septa (Figure 2H).

BDNF immunolocalization was observed in the pancreas of both species. The labeling was always observed in the cytoplasmic compartment of the positive cells. In *Otaria flavescens,* numerous disseminated moderate immunoreactive cells (ic) were observed in the exocrine pancreas (Figure 3A). In the endocrine component, strong BDNF-ic were distributed in large (Figure 3B) and medium islets with elongated shapes (Figure 3C). Moreover, the presence of positive fibers along the wall of the blood vessels and a few fibers with some positive neurons in the connective septa were also observed (Figure 3D).

In the exocrine component of *Tursiops truncatus*, moderate BDNF immunolocalization was displayed in numerous disseminated cells in the exocrine acini (Figure 3E). In the endocrine component, however, strong positive cells were detected in some islets of Langerhans (Figure 3F). Moreover, immunopositivity was observed in the walls of blood vessels (Figure 3G) and fibers (Figure 3H).

## 4. Discussion

Herein, we investigated the presence of the neurotrophins NGF and BDNF in the pancreas of two species of marine mammals, *Otaria flavescens* (South American sea lion) belonging to the order of the Carnivora and *Tursiops truncatus* (common bottlenose dolphin) belonging to the order of the Artiodactyla.

From the morphological standpoint, the pancreas of the South American sea lion appears prevalently characterized by very large islets with elongated shapes and a few small ovoid-shaped islets. It more resembles that of the bear seal and is thus closer to the terrestrial canine organ [46]. Whereas the pancreas of bottlenose dolphins is mainly organized in several ovoid shape islets of medium and small size, resembling more than that of terrestrial ungulates [47,48]. This is in line also with the different organization of the gastro-enteric apparatus of the two species. In South American sea lions, the gastroenteric apparatus resembles more that of carnivores [49,50], with a unique stomach chamber and a small and large intestine including the cecum [50]. Whereas, the common bottlenose dolphin possesses a polycamerate stomach [42,50] and an intestine subdivision into an anterior and posterior [51]. Notably, the cecum is lacking [50]. Our morphological observations thus support the theory proposed by Colegrove and collaborators [52], according to which the architecture of the islets of Langerhans in marine mammals is preserved, with regard to evolutionary adaptation to a common marine environment and diet.

With regard to the immunolocalization of NGF and BDNF, the two neurotrophins were widely distributed in both the exocrine and endocrine components of the pancreas of the two species. It is widely acknowledged that NGF provides trophic support to neurons and promotes neuritis growth [1]. In the pancreas, this neurotrophin is an important regulator for the synthesis and secretion of epidermal growth factor and insulin from the β-cells [53] by modulating electrical activity [17]. Moreover, Houtz and coworkers demonstrated that glucose homeostasis relies on NGF derived from pancreatic vascular contractile cells, where this peptide was robustly expressed [54]. Different studies have shown that the intricate relationship between vascular cells and pancreatic β-cells plays a crucial role in both islet development and adult islet function through the secretion of growth factors and other molecules [55]. Hence, the immunoreactivity of NGF along the walls of the blood vessels of the pancreas of both species makes us hypothesize that there is a regulatory mechanism similar to terrestrial mammals [56].

The presence of BDNF in the pancreas of both test species may reflect the re-known role of this molecule in regulating glucose metabolism and controlling appetite and body weight through the modulation of insulin and glucagon secretion in response to changes in plasma glucose levels as in terrestrial mammals [19,23,24]. In particular, the ability of BDNF to improve insulin sensitivity and influence energy expenditure [57] suggests that it could help mitigate the effects of postprandial hyperglycemia and contribute to the lack of ketosis during fasting periods in bottlenose dolphins. In South American sea lions, BDNF could have a key role in modulating the metabolic response during long periods of fasting, helping to optimize the use of energy reserves and maintain the functionality of the nervous system [58]. Very interestingly, BDNF was observed in the basal lamina of vessels and in neurons dispersed in the pancreatic parenchyma of the two species. The influence of BDNF on the endothelial cells is well known, impacting vessel integrity and increasing vascular permeability in particular, affecting the passage of molecules across endothelial barriers [59].

## 5. Conclusions

This preliminary study provides evidence of the presence and distribution of NGF and BDNF in the pancreas of the South American sea lion *Otaria flavescens* and common bottlenose dolphins *Tursiops truncatus*, opening up new perspectives on the understanding of the anatomy and physiology of these species and the adaptive mechanisms to their ecological niche. Further studies are needed to explore the functional role of neurotrophins in the pancreas of marine mammals, including their interaction with specific receptors and the effect on endocrine and metabolic regulation on the one hand and the comparison of their distribution between healthy and diseased animals on the other.

In conclusion, this study enriches our understanding of the presence and distribution of neurotrophins in marine mammals, suggesting potential parallels with the physiology of terrestrial mammals.

## Figures and Tables

**Figure 1 animals-14-02336-f001:**
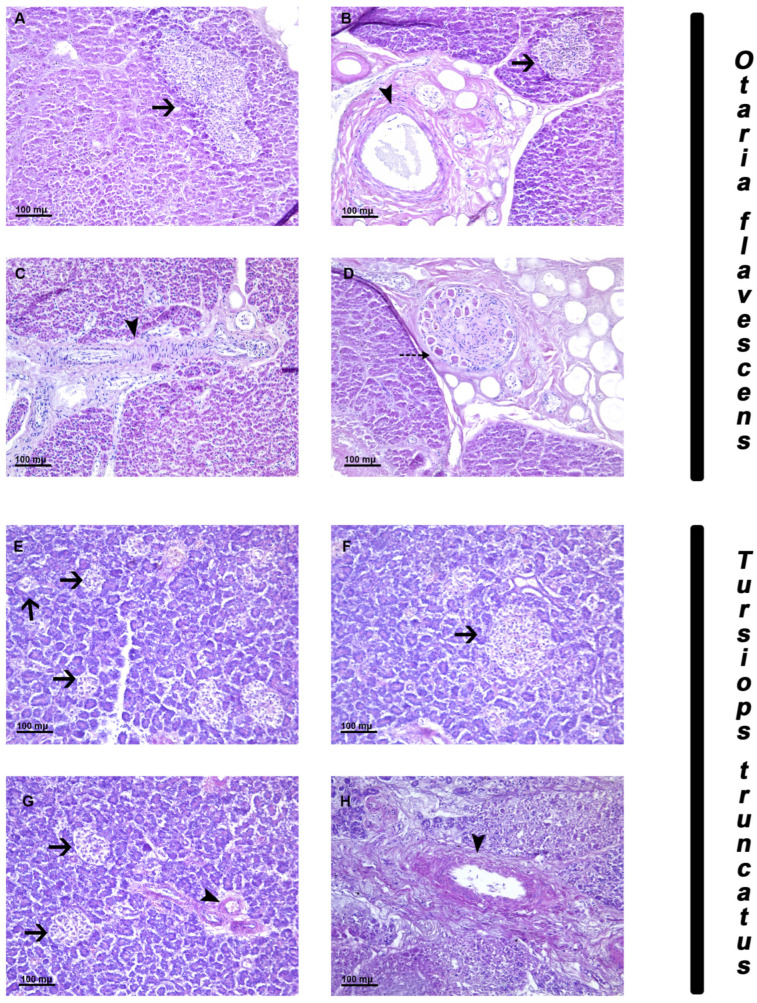
Hematoxylin eosin staining of the pancreas of *Otaria flavescens* and *Tursiops truncatus*. (**A**–**D**) *Otaria flavescens.* Cells in the parenchyma and large elongate islet (arrow) (**A**); Langerhans islets of medium size and ovoid shape (arrow) and blood vessels (arrowhead) (**B**); blood vessels in the interconnective septa (arrowhead) (**C**); ganglion with neurons in the parenchyma (dotted arrow) (**D**). (**E**–**H**) *Tursiops truncatus.* Exocrine cells in the parenchyma and islet of Langerhans of small size with ovoid shape (arrow) (**E**); islets of Langerhans of medium size (arrow) (**F**); islet of Langerhans small size (arrow) and blood vessels (arrowhead) (**G**); blood vessels (arrowhead) (**H**). Scale bar 100 µm.

**Figure 2 animals-14-02336-f002:**
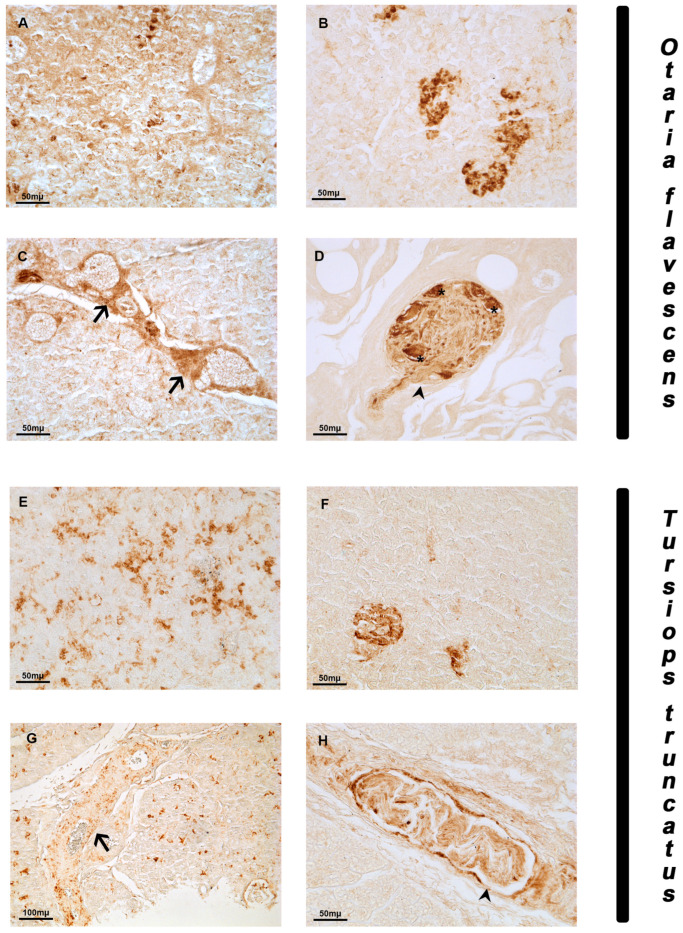
NGF immunoreactivity in the pancreas of *Otaria flavescens* and *Tursiops truncatus*. (**A**–**D**) *Otaria flavescens*. Positive exocrine cells (**A**); positive islets of Langerhans (**B**); positive fibers in the vessel wall (arrow) in the interconnective septa (**C**); ganglion with positive neurons (asterisk) and fibers (arrowhead) (**D**). (**E**–**H**) *Tursiops truncatus.* Positive exocrine cells in parenchyma (**E**); small islets of Langerhans (**F**); immunoreactivity in fibers along blood vessels wall (arrow) (**G**); positive fibers bundle (arrowhead) (**H**). Scale bar 100 µm.

**Figure 3 animals-14-02336-f003:**
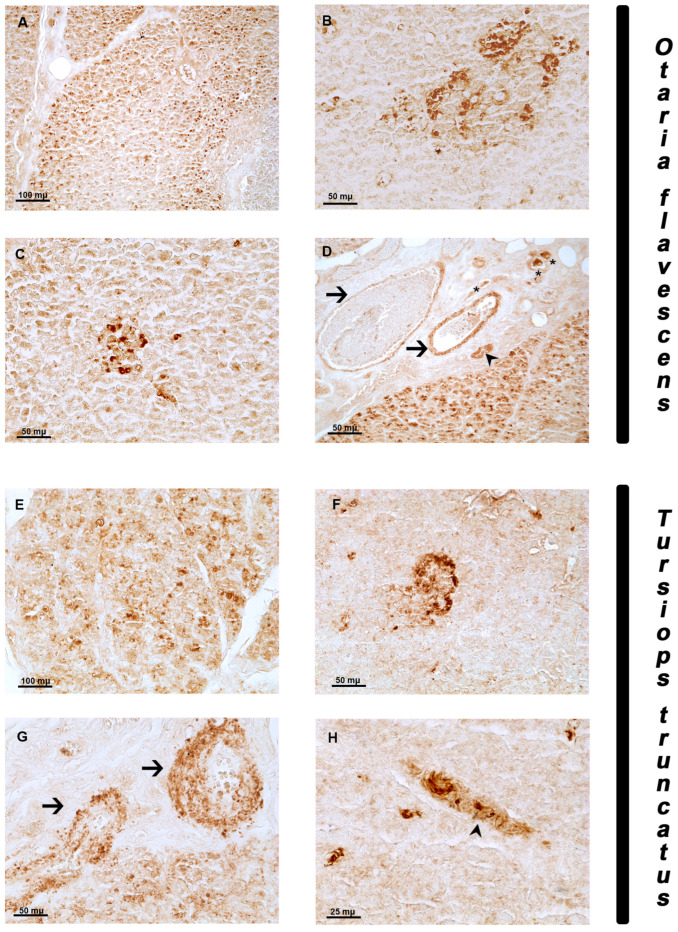
BDNF immunoreactivity in the pancreas of *Otaria flavescens* and *Tursiops truncatus*. (**A**–**D**) *Otaria flavescens*. BDNF immunorectivity of exocrine cells in parechima (**A**); positive large islet of Langerhans (**B**); positive medium islet of Langerhans (**C**); positive fibers in the wall of blood vessels (arrow), positive neurons (asterisk) and fibers (arrowhead) in the parenchyma (**D**); (**E**–**H**) *Tursiops truncatus.* Immunoreactivity of exocrine cells in parenchyma (**E**); positive islet of Langerhans (**F**); positive fibers in the wall of blood vessels (arrow) (**G**); immunoreactive fibers (arrowhead for fiber bundle) (**H**). Scale bar 100 µm.

## Data Availability

All data generated or analyzed during this study are included in this published article.

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
