# Peer review of "Immunolocalization of Two Neurotrophins, NGF and BDNF, in the Pancreas of the South American Sea Lion *Otaria flavescens* and Bottlenose Dolphin *Tursiops truncatus"

_animals, 2024, doi:10.3390/ani14162336_

Round 1

Reviewer 1 Report

Comments and Suggestions for Authors

The study reports for the first time the presence of two neurotrophins in two species of marine mammals. The manuscript is well-written, concise, and supported by appropriate references. I congratulate the authors for their work and enjoyed reading it. I have only minor comments:

I suggest including in the title: "two neurotrophins (NGF and BDNF)" to make it more understandable for non-specialist readers.

The correct species name is Otaria flavescens as demonstrated by Rodriguez and Bastida (1993). Despite the Marine Mammal Science Committee's change to byronia, we believe the validity of the senior name Otaria flavescens (Shaw 1800) is supported by nomenclatural, taxonomic, and usage arguments. Phoca flavescens Shaw is an available and priority name over Phoca byronia de Blainville under the provisions of the International Code of Zoological Nomenclature. No nomenclatural act suppressed Phoca flavescens in favor of Phoca byronia as the valid name applied to the Southern sea lion, and the Opinion 1962 of the International Commission on Zoological Nomenclature did not refer to and has accordingly no effect on Phoca flavescens, because it only rules on the availability of Phoca byronia de Blainville as a junior synonym of Phoca leonina Molina 1782. The inclusion of Otaria byronia in the Official List of Specific Names in Zoology gives it neither priority nor validity, since according to the Code this list is composed of available names, whether valid or invalid. Regarding the taxonomical identification of Shaw's lost holotype (a two feet long yellowish pup collected in the Strait of Magellan), three revisions—including the First Reviser—provided tangible information to confirm that Phoca flavescens size, color, and ear size corresponded to those of a newborn Southern sea lion pup, and not to a South American fur seal (the only other Otariid breeding near the type locality). Those authors who consider the holotype as unidentifiable did not present tangible information to refute the correspondence of the holotype traits with those found in Otaria pups. Finally, the use of O. flavescens is more frequent than that of O. byronia in the scientific literature (ca. 80%; database by Rodríguez et al. [2021] is available at https://doi.org/10.5281/zenodo.5635390) and is the official name of the species in all countries where it is distributed. We understand there are strong arguments supporting the validity and use of Otaria flavescens (Shaw, 1800) for the Southern sea lion.

The order of the South American sea lion, as well as all pinnipeds, is Carnivora, and the order of cetaceans is Order Artiodactyla, Infraorder Cetacea. This should be corrected in both the introduction and the discussion.

In line 125, it states that "Positive controls were performed by sections of zebrafish taste buds." Could you elaborate on why a fish was used as a control instead of a terrestrial mammal?

Author Response

Reviewer: 1

The study reports for the first time the presence of two neurotrophins in two species of marine mammals. The manuscript is well-written, concise, and supported by appropriate references. I congratulate the authors for their work and enjoyed reading it.

  • We are grateful to the Reviewer for the kind comment Please find the point-by-point reply as follows:

I suggest including in the title: "two neurotrophins (NGF and BDNF)" to make it more understandable for non-specialist readers.

  • We changed the title as suggested.

The correct species name is Otaria flavescens as demonstrated by Rodriguez and Bastida (1993). Despite the Marine Mammal Science Committee's change to byronia, we believe the validity of the senior name Otaria flavescens (Shaw 1800) is supported by nomenclatural, taxonomic, and usage arguments. Phoca flavescens Shaw is an available and priority name over Phoca byronia de Blainville under the provisions of the International Code of Zoological Nomenclature. No nomenclatural act suppressed Phoca flavescens in favor of Phoca byronia as the valid name applied to the Southern sea lion, and the Opinion 1962 of the International Commission on Zoological Nomenclature did not refer to and has accordingly no effect on Phoca flavescens, because it only rules on the availability of Phoca byronia de Blainville as a junior synonym of Phoca leonina Molina 1782. The inclusion of Otaria byronia in the Official List of Specific Names in Zoology gives it neither priority nor validity, since according to the Code this list is composed of available names, whether valid or invalid. Regarding the taxonomical identification of Shaw's lost holotype (a two feet long yellowish pup collected in the Strait of Magellan), three revisions—including the First Reviser—provided tangible information to confirm that Phoca flavescens size, color, and ear size corresponded to those of a newborn Southern sea lion pup, and not to a South American fur seal (the only other Otariid breeding near the type locality). Those authors who consider the holotype as unidentifiable did not present tangible information to refute the correspondence of the holotype traits with those found in Otaria pups. Finally, the use of O. flavescens is more frequent than that of O. byronia in the scientific literature (ca. 80%; database by Rodríguez et al. [2021] is available at https://doi.org/10.5281/zenodo.5635390) and is the official name of the species in all countries where it is distributed. We understand there are strong arguments supporting the validity and use of Otaria flavescens (Shaw, 1800) for the Southern sea lion.

  • We changed Otaria byronia in Otaria flavescens according to the comments.

The order of the South American sea lion, as well as all pinnipeds, is Carnivora, and the order of cetaceans is Order Artiodactyla, Infraorder Cetacea. This should be corrected in both the introduction and the discussion.

  • We made the corrections.

In line 125, it states that "Positive controls were performed by sections of zebrafish taste buds." Could you elaborate on why a fish was used as a control instead of a terrestrial mammal?

  • We thank the reviewer for the observation regarding immunohistochemistry control. Neurotrophins are evolutionary well conserved proteins from fish to mammals, despite the duplication events which have led to have an additional neurotrophin in teleost fish (doi: 10.1002/cne.23457; doi: 10.1002/cne.24391), both in terms of aminoacid sequences and neurotrophic action. Remarkably the two antibodies employed in this study recognize a conserved epitope both in mammals and fish, as previously reported [44]. In light of this, we decided to use fish tissues among our experimental controls.  

Reviewer 2 Report

Comments and Suggestions for Authors

The paper submitted by the authors provides relevant insights about the localization of the neurotrophins in the pancreas of bottlenose dolphins and sea lions, as well as the morphology and distribution of the islets in these two species. Before acceptance, I recommend to check the comments listed below.

Lines 58-63, please rewrite since it is a little confusing.
Lines 68-69, this paragraph is unclear, please redo.
Line 82, please add a reference.
Line 104, here, you do not provide information about the health status of the animals taking in consideration that live stranded animals shows stress conditions associated with elevation of fibrinogen and alpha 1 anti trypsin  that can affect glucose levels.

Line 107, please change “morphological evaluation at optic level” for “histological evaluation”.

Line 114, please add the country after “biomedical”.

Line 115, please change (1:200) by diluted at…

Figure 1, please add arrows pointing the blood vessels. In addition, it should be more appropriate to show a picture with islets of different sizes.

Figures 2-3, here it would be more illustrative to add arrows labeling specific findings explaining along the text (fibers, neurons…). In addition, most of the IHC images show a lot of background and noise. Did you check different dilutions? It would be interesting to reduce them in order to better understanding of the images.

Lines 159-164, please explain how was the immunoreactivity, strong or weak affecting high or low numbers of cells? Citoplasmic or nuclear?

Lines 235-236, this sentence is a little redundant, please check.

In the conclusion section, I miss a sentence including “comparing their distribution between normal animals and diseased ones”

Author Response

Reviewer: 2

The paper submitted by the authors provides relevant insights about the localization of the neurotrophins in the pancreas of bottlenose dolphins and sea lions, as well as the morphology and distribution of the islets in these two species.

  • We are grateful to the Reviewer for the kind comment Please find the point-by-point reply as follows:

Lines 58-63, please rewrite since it is a little confusing.

  • The text was changed as follows: “BDNF not only serves as a powerful pro-survival factor for neurons in both the central and peripheral nervous system and a strong modulator of brain synaptic plasticity [18], but it also plays a crucial role in the regulation of body weight, energy expenditure and glucose metabolism [19]. When brain BDNF levels are depleted, it can lead to excessive nutrition and increased body weight, which are often accompanied by metabolic syndrome characterized by hyperleptinemia, hyperglycemia and hyperinsulinemia [20].”

Lines 68-69, this paragraph is unclear, please redo.

  • The text was changed as follows: “Moreover, BDNF enhances insulin sensitivity in peripheral tissues, boosts energy expenditure in db/db mice and reduce their food intake [23-24]”.

Line 82, please add a reference.

  • We added the following references:

Davis R.W. Marine mammals. Adaptations for an aquatic life. 1st ed. Springer Cham. Springer Nature Switzerland, 2019; pp. 1-302.

Reidenberg, J.S. Anatomical adaptations of aquatic mammals. Anat Rec (Hoboken) 2007, 290(6), 507–513.

Line 104, here, you do not provide information about the health status of the animals taking in consideration that live stranded animals shows stress conditions associated with elevation of fibrinogen and alpha 1 anti trypsin that can affect glucose levels.

  • We took the reviewer point which is relevant in the descriptive study of comparative anatomy, especially when compared to a pathological anatomy investigation. As mentioned, the tissues were provided by the Mediterranean Marine Mammal Tissue Bank (MMMTB), and no pathological remarks were referred to such samples, although we could not exclude that subclinical pathological conditions could affect these stranded animals. Nevertheless, our short report is exclusively addressed to evaluate the immunodistribution of NGF and BDNF in the pancreas of the two species, and we cared of the good tissue preservation in order to ensure repeatable experimental immunohistochemical trials. Investigating the potential correlations with the physiological glucose levels represent the future scientific perspective (doi: 10.3389/fphys.2018.01845) [Ref - Frontiers in Physiology] to better disentangle the role of the two neurotrophins in the metabolic regulation of two species with different feeding habits.

Line 107, please change “morphological evaluation at optic level” for “histological evaluation”.

  • We modified the text accordingly.

Line 114, please add the country after “biomedical”.

  • We added it.

Line 115, please change (1:200) by diluted at…

  • We modified the text accordingly.

Figure 1, please add arrows pointing the blood vessels. In addition, it should be more appropriate to show a picture with islets of different sizes.

  • We added in Fig 1 B, C, G and H arrowheads indicating blood vessels.

In Figure 1 islets of different sizes were shown for both species.

Figures 2-3, here it would be more illustrative to add arrows labeling specific findings explaining along the text (fibers, neurons…). In addition, most of the IHC images show a lot of background and noise. Did you check different dilutions? It would be interesting to reduce them in order to better understanding of the images.

  • We modified Figure 2-3 labeling specific findings: Fig 2C arrow for fibers along vessel walls; Fig 2D arrowhead for ganglion positive fibers and asterisk for positive neurons in ganglion; Fig 2G arrow for fibers along vessels walls; Fig 2H arrowhead for positive fiber bundle; Fig 3H arrowhead for fiber bundle.

We agree that in some images there is a prevalent background or noise, which may cofound the immunostaining. This is for example the case of immunodistribution in the exocrine components of the pancreas (Fig. 2A), differently from the endocrine component (Fig. 2B) where this effect is much blunt. Therefore, the tested dilution was the only which allowed to obtain a good signal in both exocrine and endocrine components.

Lines 159-164, please explain how was the immunoreactivity, strong or weak affecting high or low numbers of cells? Citoplasmic or nuclear?

  • We modified the text to better explain the immunoreactivity.

For both neurotrophins, we detected cytoplasmatic signal and we have added this specification in the main text.

Lines 235-236, this sentence is a little redundant, please check.

  • We deleted the sentence “being this molecule involved in the regulation of body weight and energy expenditure”.

In the conclusion section, I miss a sentence including “comparing their distribution between normal animals and diseased ones”

  • We added it.
